# Visual-Language Collaborative Representation Network for Broad-Domain Few-Shot Image Classification

Submission Id: 1441*

## ABSTRACT

Visual-language models based on CLIP have shown remarkable abilities in general few-shot image classification. However, their performance drops in specialized fields such as healthcare or agriculture, because CLIP's pre-training does not cover all category data. Existing methods excessively depend on the multi-modal information representation and alignment capabilities acquired from CLIP pre-training, which hinders accurate generalization to unfamiliar domains. To address this issue, this paper introduces a novel visual-language collaborative representation network (MCRNet), aiming at acquiring a generalized capability for collaborative fusion and representation of multi-modal information. Specifically, MCRNet learns to generate relational matrices from an information fusion perspective to acquire aligned multi-modal features. This relationship generation strategy is category-agnostic, so it can be generalized to new domains. A class-adaptive fine-tuning inference technique is also introduced to help MCRNet efficiently learn alignment knowledge for new categories using limited data. Additionally, the paper establishes a new broad-domain few-shot image classification benchmark containing seven evaluation datasets from five domains. Comparative experiments demonstrate that MCRNet outperforms current state-of-the-art models, achieving an average improvement of 13.06% and 13.73% in the 1-shot and 5-shot settings, highlighting the superior performance and applicability of MCRNet across various domains.

## CCS CONCEPTS

• **Computing methodologies → Activity recognition and understanding**.

## KEYWORDS

Visual-language modeling, Representation learning, Few-shot image classification

**ACM Reference Format:**

Anonymous Author(s). 2024. Visual-Language Collaborative Representation Network for Broad-Domain Few-Shot Image Classification. In *Proceedings of Proceedings of the 32th ACM International Conference on Multimedia (MM '24)*. ACM, New York, NY, USA, 10 pages. https://doi.org/XXXXXXX.XXXXXXX

## 1 INTRODUCTION

Few-shot image classification (FSIC) is a fundamental task in computer vision that has garnered widespread attention in recent years [38, 40, 42]. FSIC aims for models to acquire meta-knowledge from a large number of base classes and subsequently adapt rapidly to novel classes using a few support images, enabling the classification of query images. Numerous visual few-shot learning (FSL) models based on meta-learning [12, 23, 25, 30] or metric learning [17, 39, 44, 48] have been employed to tackle FSIC, but they have not yielded satisfactory performance. Recently, the emergence of CLIP [34] presents a new multi-modal view to address FSIC. Based on CLIP, the state-of-the-art (SOTA) visual-language methods (VLMs) [14, 51, 53] have showcased impressive performance on general domain datasets, achieving over 60% accuracy on ImageNet [8] and 90% accuracy on Caltech-101 [11] with only one support image.

However, when existing VLMs are applied to specific fields such as medicine [33, 45], agriculture [24], or industry [15], their performance is less than ideal. This deficiency arises from the challenge that CLIP's pre-training classes are difficult to encompass all categories across various domains, leading to its limited capability in representing and aligning images and text from unfamiliar categories. Existing works focus on enhancing CLIP by designing new text prompts or adapters, yet they fail to address CLIP's poor generalization when confronted with unfamiliar domains. For example, in fine-grained butterfly classification, as shown in Fig. 1 (a) and (b), CLIP struggles to accurately extract information from new categories such as "cabbage butterfly" or "pachliopta aristolochiae butterfly", resulting in biased matching computations. Other models based on CLIP fail to align features of images and text from new classes, hence yielding inaccurate results.

To address the issue, this paper introduces a novel visual-language collaborative representation network (MCRNet) that aims to learn a generalized capability for aligning and representing multi-modal information. As depicted in Fig. 1 (c), MCRNet comprises three components: visual-text encoders, a collaborative relation learner, and a multi-feature re-presentation module. The visual-text encoders are used to extract prototype features of multi-modal information. The collaborative relation learner extensively interacts with prototype features of support or query images and texts and generates relationship matrices for support or query. The re-presentation module recalculates the prototype features and relationship matrices through weighted computations. What sets it apart from existing methods is that MCRNet coordinates the representation and fusion processes of visual and language modalities, which strengthens the alignment between multi-modal information by learning relationships among different modal semantics. This ability to generate relationships between different modalities to enhance representation can be widely applied in various tasks, allowing MCRNet to

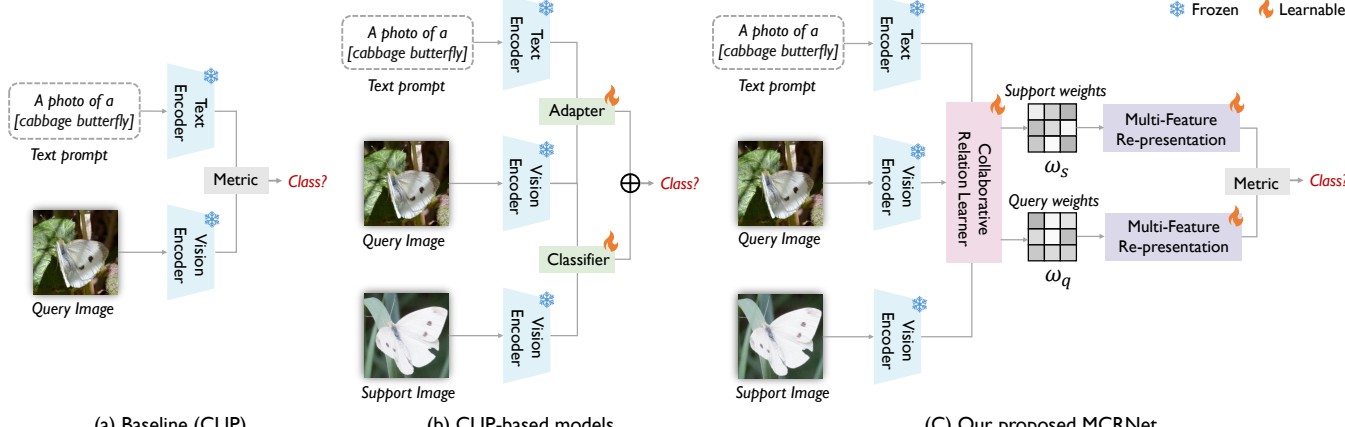

**Figure 1: The comparison of framework between (a) the baseline CLIP [34], (b) CLIP-based visual-language models, and the (c) visual-language collaborative representation network (MCRNet) proposed in this paper.**

adapt more quickly to new category knowledge. To better apply to new category tasks, a class-adaptive fine-tuning inference method is also proposed, aiming to rapidly learn from the given support data and save testing time under the $n$-support images scenario.

Furthermore, this paper establishes a new broad-domain few-shot image classification (BD-FSIC) benchmark to evaluate the generalization and applicability of existing models across a wide range of domain tasks. This benchmark comprises seven evaluation datasets covering five domains including biology, agriculture, medicine, mining industry, and archaeology. Finally, the evaluation studies on the BD-FSIC benchmark validate that existing visual-language models exhibit notably lower performance on unfamiliar domain tasks when compared to general datasets. The comparative experiments show that MCRNet surpasses the current SOTA models, achieving an average improvement of 13.06% in the 1-shot scenario and 13.73% in the 5-shot scenario, outperforming both existing visual-language models and visual few-shot learning models. These results demonstrate the superiority of MCRNet, as well as its transferability and generalizability across multi-domain tasks. In summary, this paper has the following contributions:

- This paper introduces a novel multi-modal collaborative representation network (MCRNet) to enhance the alignment and representation capabilities of multi-modal information in unfamiliar domains. Unlike existing methods, MCRNet adopts a universal relational matrix learning approach to facilitate the fusion and feature representation of multi-modal information.
- This paper constructs a new broad-domain few-shot image classification benchmark comprising seven evaluation datasets spanning five domains.
- Comparative experiments demonstrate that MCRNet outperforms the existing SOTA methods by over 12% average accuracy on seven evaluation datasets across multiple settings, showcasing the advancement and domain applicability of MCRNet.

## 2 RELATED WORK

### 2.1 Few-Shot Learning

Visual few-shot learning (FSL) is the primary approach used to address few-shot image classification tasks. Existing FSL models mainly focus on a purely visual perspective and can be categorized into three types. The first type involves prototype representation learning [6, 28, 35, 36, 42, 47, 49], which focuses on learning more generalizable feature representations to classify query images quickly on new categories after fine-tuning. The second type is based on metric learning [17, 39, 44, 48], which deals with how to measure feature distances in the manifold space, and this meta-ability of measurement can be transferred to new categories. The third type is based on meta-learning [12, 23, 25, 30], known as "learning to learn", where multiple different tasks are constructed during pre-training to learn a general classification ability from these tasks, which is then applied to new categories. Recently, more research has been focusing on transferring FSL models to specific domains such as healthcare [5, 16, 31] and industry [13, 21]. However, their performance decreases compared to general domains because, in specific domain applications, textual descriptions play a crucial role in capturing important information in images. Therefore, this paper aims to draw inspiration from representation learning methods in visual FSL and fully utilize the textual information provided by support to address the aforementioned challenges.

### 2.2 Vision-Language Models

In recent years, there has been a growing focus on leveraging language cues to enhance the performance of visual tasks. This has led to significant attention being drawn to vision-language models (VLMs) [26, 32, 50], which are pre-trained using a vast amount of image-text pairs readily available on the internet and can be directly applied to downstream visual tasks. In particular, the introduction of the CLIP [34] has sparked a wave of interest in using CLIP-based approaches to address basic visual tasks such as image classification or segmentation. CLIP utilizes an image-text contrastive objective, aligning paired images and texts closely while pushing others apart

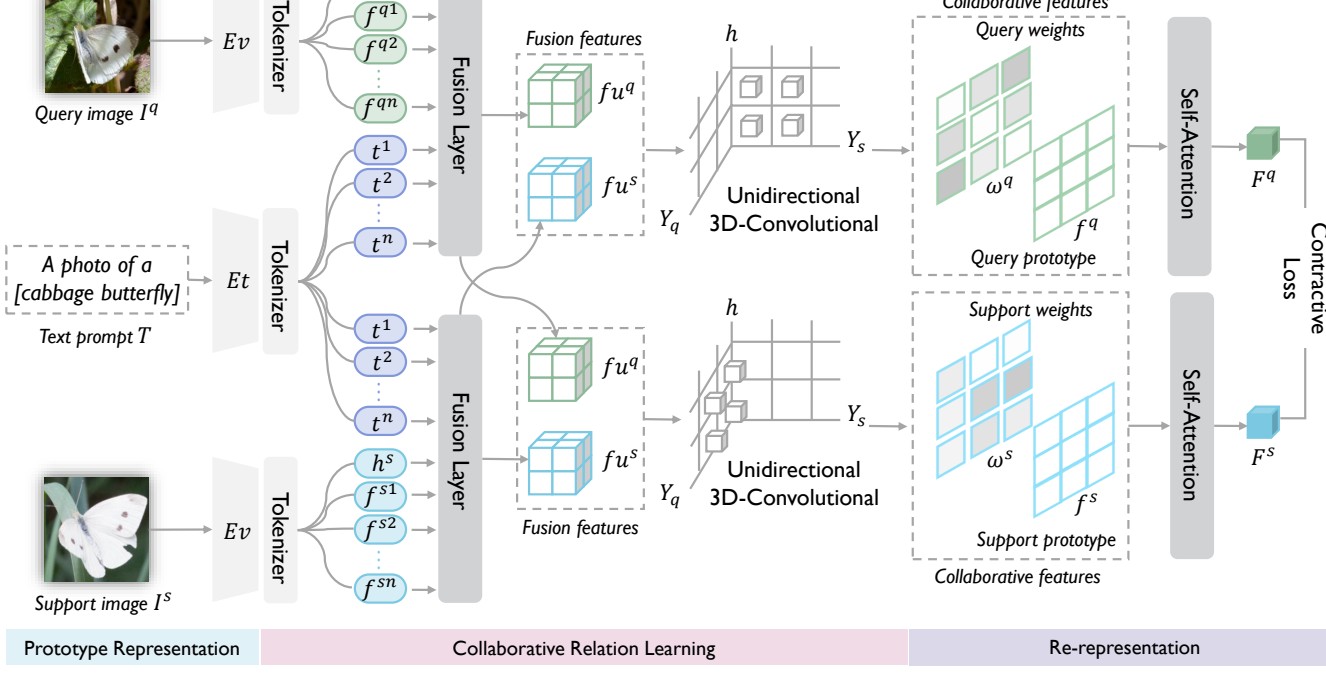

**Figure 2: The overview of visual-language collaborative representation network (MCRNet). MCRNet consists of three parts: prototype representation, collaborative relation learning, and re-representation. The entire network is supervised by the contrastive loss. $E_v$ and $E_t$ refer to the visual and text encoders of CLIP respectively.**

in the embedding space. This approach allows pre-trained VLMs to capture intricate vision-language correspondence knowledge. Subsequently, CLIP-based models [14, 51–53] have been proposed to enhance CLIP's performance in few-shot image classification. These models primarily achieve this by designing adapters that can be quickly fine-tuned or by incorporating methods such as image matching. They leverage the multi-modal information features generated by CLIP to compute similarities between the test image and prompt text or between the test image and known images to determine the category of the test image. However, excessive reliance on features generated by CLIP can lead to representation biases when the model encounters text and images of categories that were rarely seen or unseen during pre-training. This bias in representations can result in classification errors when further matching is performed based on these representations. To improve the performance of existing VLMs in unfamiliar domains or categories, such as fine-grained butterfly classification in the field of biology or plant virus classification in agriculture, this study proposes a novel visual-language cooperative representation method to learn a highly generalizable multi-modal information representation for multi-domain few-shot image classification.

## 2.3 Related Datasets

FSL models and VLMs are evaluated on general domain benchmarks such as miniImageNet [20], CIFAR [11], CUB [9], or tiered-ImageNet [11]. However, there is a scarcity of evaluation datasets specific to domains. Despite the introduction of a cross-domain

few-shot learning benchmark by Guo Y et al. [18], which only includes two datasets from the medical domain and one from the agricultural domain. To bridge this gap, this study establishes a new benchmark encompassing five domains with seven datasets, to offer a comprehensive platform for assessing model transferability and generalization across diverse domains in image classification.

## 3 METHOD

### 3.1 Problem Formulation

In broad-domain few-shot image classification (BD-FSIC), we adhere to the classic FSIC problem setting, where the model is pre-trained on a large-scale base class dataset $C_{\text{base}}$ and then evaluated on novel classes $C_{\text{novel}}$ in unfamiliar domains. Both training and evaluation are conducted in $N$-way-$K$-shot episodes [37]. Specifically, let the $\mathcal{D}_{train} = \left\{ \left( I_i^q, y_i^q, \left\{ I_i^{sk}, T_i^s, y_i^{sk} \right\}_{k=1}^K \right) \right\}_{i=1}^{N_t}$ represent $N_t$ training episodes from $C_{base}$ and $K$ refers to the $K$-th sample of class $N$. Here, $I_i^q$ represents the query image and $y_i^q$ represents the corresponding class label. $I_i^{sk}$ represents the support image, $T_i^s$ represents the text prompt of support image, and $y_i^{sk}$ represents the class label. In each training episode, K images $I_i^{sk}, T_i^s$, and their corresponding labels $y_i^{sk}$ are sampled from each of the randomly selected N classes to form the support set. Additionally, other images $I_i^q$ are sampled from these classes to form the query set, and $y_i^q$ is used as supervision to optimize the model. For evaluation,

let $\mathcal{D}_{test} = \left\{ \left( I_i^q, \left\{ I_i^{s_k}, T_i^s, y_i^{s_k} \right\}_{k=1}^K \right) \right\}_{i=1}^{N_e}$ represent the test episodes. $N_t$ is from $C_{\text{novel}}$. The model predicts the class $\hat{y}_i^q$ of $I_i^q$ based on $I_i^{s_k}, T_i^s$, and $y_i^{s_k}$, compares it with the true label $y_i^q$, and calculates the model's accuracy. Hence, for visual FSL models and VLMs, the key to addressing BD-FSIC lies in learning more generalized representations or aligning meta-knowledge from $C_{\text{base}}$ and effectively leveraging the relevant information from $I_i^{s_k}$ in $C_{\text{novel}}$.

## 3.2 Overview

The proposed MCRNet is designed to achieve feature alignment with generalization capabilities through a collaborative fusion and representation method for vision and text. As illustrated in Fig. 2, MCRNet consists of three components: the prototype representation based on CLIP, the collaborative relation learning, and the re-representation part. The prototype representation comprises text and visual encoders from CLIP, mapping input images and text information into prototype features. The collaborative relation learner, including a fusion layer and two unidirectional 3D-convolutional layers, fuses visual-text prototype features and learns the relationship matrix between support and query information. The multi-feature re-presentation learner incorporates a simple self-attention layer to relearn the alignment of the relationship matrix and prototype features for a refined feature representation. The entire network is supervised by a contrastive learning loss, aiming to bring similar support and query features closer while pushing different-class features apart. Additionally, a category-adaptive fine-tuning method was proposed to assist MCRNet in rapid learning on limited data.

## 3.3 Collaborative Relation Learner

During each training or testing episode, we acquire support images $I_i^s$ and query images $I_i^q$ along with support category textual descriptions $T_i^s$. If $I_i^s$ and $I_i^q$ belong to the same category, they form a positive sample pair; if they belong to different categories, they form a negative sample pair. Subsequently, through the image encoder $E_v$ and text encoder $E_t$ of CLIP, the aforementioned multi-modal information is mapped to $f_i^s \in \mathbb{R}^{W \times H \times C}$, $f_i^q \in \mathbb{R}^{W \times H \times C}$, and $t_i^s \in \mathbb{R}^{M \times 1}$. These features are referred to as prototype features, As in Fig. 2 (Prototype Representation).

Subsequently, as illustrated in Fig. 2 (Collaborative Relation Learning), MCRNet merges the prototype features and generates a multi-modal information relational matrix. The first step involves multi-modal information fusion. The textual prototype feature $t_i^s$ obtained is fused and computed with the image prototype features of support $f_i^s$ and query $f_i^q$. Specifically, initially, these features are tokenized, transforming $f_i^s$ and $f_i^q$ dimensions to $\mathbb{R}^{(W \times H) \times C}$ and adding position embedding. Subsequently, $t_i$ is concatenated with $f_i^s$ and $f_i^q$ to obtain the fused feature tokens $u_i^s$ and $u_i^q$. Following this, an attention mechanism is utilized, defined as:

$$Attention(Q, K, V) = Softmax\left( \frac{QK^T}{\sqrt{d}} \right) V. \tag{1}$$

We perform self-attention calculations on $u_i^s$ and $u_i^q$ separately. Since the fused features include both image and text features, important category features in the fused features are assigned higher

weights during multiple similarity calculations. This is expressed as:

$$fu_i = Attention(u_i W_\phi^Q, u_i W_\phi^K, u_i W_\phi^V), \tag{2}$$

where $W_\phi^Q, W_\phi^K, W_\phi^V$ are learnable weights with a size of $d \times d$. The resulting features are then normalized and mapped back to the original $\mathbb{R}^{(W \times H) \times C}$ through a linear layer, obtaining the initialized fused features $fu_i^s$ and $fu_i^q$.

After obtaining $fu_i^s$ and $fu_i^q$, we proceed with the subsequent relational matrix generation process. Initially, we reshape $fu_i^s$ and $fu_i^q$ to $\mathbb{R}^{W \times H \times C}$, then vertically concatenate the two three-dimensional feature matrices to form $fu_i \in \mathbb{R}^{W \times H \times 2 \times C}$. This means that the fused features of support and query are stored without compression in $fu_i$. Afterward, we designed a unidirectional 3D-convolutional process to compress and learn from $fu_i$. The aim is to compress from the direction of $fu_i^q$ to $fu_i^s$ within $fu_i$ for $fus_i$ specifically. The purpose of this convolutional compression is to gradually map the information from $fu_i^q$ into $fu_i^s$, thereby generating a relational matrix $\omega^s$ in $fu_i^s$ that maximizes the similarity with $fu_i^q$. Conversely, for $fu_i^q$, a reverse convolutional compression is performed to generate $\omega^q$. When learning the relational matrix $\omega^s$ for $fu_i^s$, we assume that $L \times M \times N$ is the shape of the 3D convolutional kernel. $W$ represents the weight at position $l, m, n$ of the kernel, and we set $N$ to 1. When learning the relational matrix $\omega^q$ for $fu_i^q$, we set $L$ to 1. The specific operations are as follows:

$$\omega^s = F\left( \sum_c^C \sum_{n^l=0}^1 \sum_{w^l=0}^W \sum_{h^l=0}^H W_{(h^l, w^l, n^l)} \right. \\ \left. fu_{c,(l+h^l),(m+w^l),(n+n^l)} + b_{(h^l, w^l, n^l)} \right), \tag{3}$$

$$\omega^q = F\left( \sum_c^C \sum_{n^l=1}^2 \sum_{w^l=0}^W \sum_{h^l=0}^H W_{(h^l, w^l, n^l)} \right. \\ \left. fu_{c,(l+h^l),(m+w^l),(n+n^l)} + b_{(h^l, w^l, n^l)} \right), \tag{4}$$

where $F(.)$ is an activation function and $b_{(h^l, w^l, n^l)}$ is the bias of the computed feature map. Thus, Based on the collaborative relation learner, we obtain the weight relational matrix $\omega^q$ representing the impact of query fused features on support fused features, and the relational matrix $\omega^s$ representing the weighted impact of support fused features on query fused features.

## 3.4 Re-representation and Loss Function

We concatenate the obtained relational matrices $\omega^s$ and $\omega^q$ with the prototype features of support and query, $f_i^s$ and $f_i^q$ respectively. Subsequently, we pass them through a self-attention layer and a linear mapping layer to regenerate the final fused features of support and query, $F_i^s$ and $F_i^q$:

$$F_i = MLP(Attention(f_i W_\phi^Q, f_i W_\phi^K, f_i W_\phi^V)). \tag{5}$$

The resulting features are $F_i^q \in \mathbb{R}^{1 \times C}$ and $F_i^s \in \mathbb{R}^{1 \times C}$. This re-representation learner is lightweight, and designed for quick fine-tuning during the evaluation process. During the training phase, we use the contrastive loss described above to supervise the training

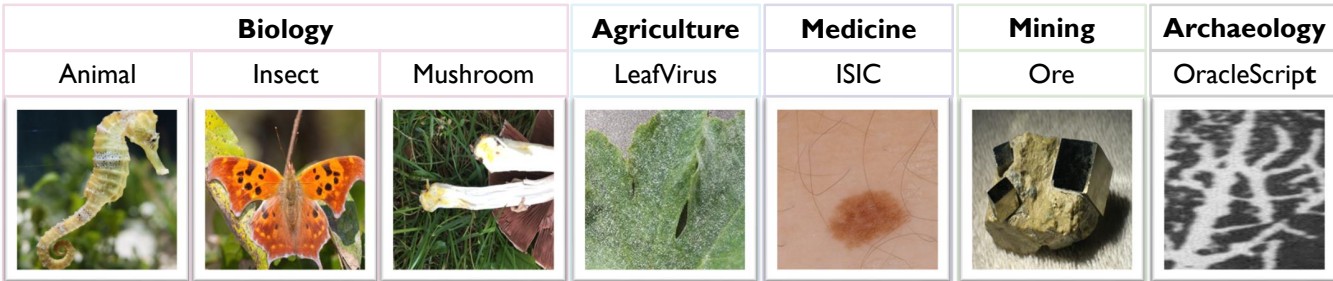

**Figure 3: This paper constructs a new broad-domain few-shot image classification (BD-FSIC) benchmark, covering five domains: biology, agriculture, medicine, mining industry, and archaeology, and encompassing seven evaluation datasets: Animal, Insect, Mushroom, Leaf Virus, ISIC, Ore. and OracleScrip.**

process beyond the prototype representation:

$$\mathcal{L}_{con} = -\frac{1}{N}\sum_{i=1}^{N}\mathbf{I}(y_i^s == y_i^q)log(d_{(Fq,Fs)}),\tag{6}$$

where $\mathbf{I}(y_i^s == y_i^q)$ indicates that if the class labels of support and query are the same, it is 1; otherwise, it is 0. The $d_{(Fq,Fs)}$ refers to the L2 distance. By reducing the distance between support and query instances of the same class and increasing the distance between instances of different classes, the goal is to enable the Collaborative Relation Learner to align multi-modal information of the same class and separate multi-modal information of different classes. This process helps in learning the final support-query relational matrix. The ability to incorporate multi-modal information of the same and different classes into the relation learning process is category-agnostic and can be generalized to new classes.

### 3.5 Class-Adaptive Fine-Tuning Inference

During the evaluation process, to fully utilize the support information, we design a fast fine-tuning method for MCRNet. Since the relational matrices learned in the collaborative relation learner exhibit strong generalization properties, we only fine-tune the Re-representation part. This is why the Re-representation learner is designed to be lightweight. Taking 5-way-5-shot as an example, where 5 classes are randomly selected from $C_{novel}$, each with five support images $I_i^s$ and $T_i^s$, we augment each $I_i^s$ into $N$ images using random rotations, cropping, and other data augmentation techniques. These augmented images are then combined with different $T_i^s$ and fed into MCRNet to obtain multiple relational matrices. By randomly combining an augmented support image with the relational matrix generated from that image, we can create multiple sets of new class data for re-representation. These class data are sequentially input into the re-representation network as either the same class or a different class to fine-tune MCRNet. This method alleviates overfitting issues caused by the limited number of support instances and helps the model learn more accurate class distributions for new classes.

### 4 THE PROPOSED BENCHMARK

This paper constructs a new broad-domain few-shot image classification benchmark (BD-FSIC), aiming to provide a comprehensive evaluation platform for existing methods in the field of image classification. As shown in Fig. 3, this benchmark covers five domains: Biology, Agriculture, Medicine, Mining, and Archaeology, encompassing evaluation datasets for seven different domain-specific tasks. The biological domain includes three datasets for different classification tasks, while each of the other domains contains one dataset. Furthermore, except for Animal, the other seven datasets are fine-grained classification datasets as these tasks are more challenging and have practical applications. Therefore, except for Animal, the other seven datasets are fine-grained. The descriptions of these datasets are provided below:

- Animal is a coarse-grained dataset containing 34 animal categories, with a total of 50, 304 images sourced primarily from the COD10K dataset [10] and collected from the web.
- Insect is a fine-grained classification dataset comprising 70 categories and 22, 242 images. It is sourced from the InsectD dataset [43] and collected from the web. Notably, the category of butterflies alone includes 18 species, challenging models to accurately differentiate between closely related categories with minimal intra-class variations.
- Mushroom consists of 51 fine-grained mushroom categories, totaling 21, 096 images sourced from AI Studio [1] and the Mushroom dataset [2, 3].
- LeafVirus is an agricultural dataset containing 6 categories of plant diseases, with a total of 1, 810 images sourced from Plant-Village [22] and AI Studio [1].
- ISIC [4, 7] consists of 2, 594 images categorized into "melanoma", "melanocytic nevus", "basal cell carcinoma", "actinic keratosis/Bowen's disease", "benign keratosis", "dermatofibroma", and "vascular lesion". It is a skin lesion classification dataset.
- Ore dataset comprises 6 different types of ores, totaling 867 images, sourced from AI Studio [1].
- OracleScript is a dataset for Oracle bone script recognition, consisting of 241 different Chinese character categories with a total of 308, 593 images sourced from [41]. Compared to MINIST [29], this dataset features more complex font characteristics, with many images having low resolutions, demanding a higher capability from models in feature extraction and matching.

**Table 1: Experimental comparison results of MCRNet and SOTA models in the biological domain (Animal, Insect, and Mushroom) as well as in the agricultural domain (Leaf Virus) on 5-way-1-shot and 5-way-5-shot settings. The numbers in bold indicate the best performance, while the underlined ones denote the second best. All the backbone of the following models is ViT.**

| Method | Animal | | Insect | | Mushroom | | LeafVirus | |
|---|---|---|---|---|---|---|---|---|
| | 1-shot | 5-shot | 1-shot | 5-shot | 1-shot | 5-shot | 1-shot | 5-shot |
| *Unimodality Few-Shot Learning Models* | | | | | | | | |
| FewTURE [CVPR2020] [46] | $34.28_{\pm0.26}$ | $44.44_{\pm0.53}$ | $32.59_{\pm0.40}$ | $44.13_{\pm0.55}$ | $29.29_{\pm0.34}$ | $43.89_{\pm0.58}$ | $53.94_{\pm0.48}$ | $74.99_{\pm0.51}$ |
| HTCTrans [CVPR2022] [20] | $42.15_{\pm0.60}$ | $47.82_{\pm0.75}$ | $\underline{47.47}_{\pm0.42}$ | $\underline{59.03}_{\pm0.63}$ | $32.71_{\pm0.29}$ | $34.17_{\pm0.52}$ | $64.87_{\pm0.55}$ | $\underline{82.92}_{\pm0.48}$ |
| CPEA [ICCV2023] [19] | $42.46_{\pm0.63}$ | $52.07_{\pm0.49}$ | $44.54_{\pm0.53}$ | $60.67_{\pm0.87}$ | $33.21_{\pm0.42}$ | $48.24_{\pm0.57}$ | $\underline{65.94}_{\pm0.39}$ | $81.54_{\pm0.55}$ |
| *Vision-Language Models* | | | | | | | | |
| CLIP [ICML2021] [34] | $73.61_{\pm0.25}$ | $74.40_{\pm0.32}$ | $20.67_{\pm0.37}$ | $20.79_{\pm0.33}$ | $45.58_{\pm0.35}$ | $46.23_{\pm0.35}$ | $35.59_{\pm0.40}$ | $34.64_{\pm0.34}$ |
| Tip-Adapter [ECCV2022] [51] | $74.06_{\pm0.47}$ | $75.38_{\pm0.49}$ | $23.10_{\pm0.56}$ | $36.68_{\pm0.53}$ | $44.25_{\pm0.46}$ | $47.99_{\pm0.41}$ | $39.91_{\pm0.44}$ | $47.24_{\pm0.55}$ |
| CoOP [CVPR2022] [52] | $\underline{75.19}_{\pm0.62}$ | $75.23_{\pm0.69}$ | $20.02_{\pm0.71}$ | $19.98_{\pm0.89}$ | $46.24_{\pm0.72}$ | $45.30_{\pm0.70}$ | $33.32_{\pm0.62}$ | $35.29_{\pm0.58}$ |
| APE-T [ICCV2023] [53] | $74.80_{\pm0.47}$ | $\underline{79.97}_{\pm0.58}$ | $21.33_{\pm0.62}$ | $21.02_{\pm0.58}$ | $48.60_{\pm0.30}$ | $48.97_{\pm0.34}$ | $39.75_{\pm0.57}$ | $41.00_{\pm0.59}$ |
| CLIP-Adapter [IJCV2024] [14] | $74.20_{\pm0.28}$ | $75.80_{\pm0.33}$ | $22.57_{\pm0.36}$ | $22.99_{\pm0.41}$ | $\underline{49.85}_{\pm0.51}$ | $52.17_{\pm0.69}$ | $36.48_{\pm0.66}$ | $37.24_{\pm0.47}$ |
| MCRNet (Ours) | $\mathbf{75.86}_{\pm0.54}$ | $\mathbf{84.33}_{\pm0.72}$ | $\mathbf{70.27}_{\pm0.76}$ | $\mathbf{81.09}_{\pm0.40}$ | $\mathbf{51.25}_{\pm0.35}$ | $\mathbf{64.97}_{\pm0.88}$ | $\mathbf{70.79}_{\pm0.68}$ | $\mathbf{88.87}_{\pm0.67}$ |

**Table 2: Experimental comparison results of MCRNet and SOTA models in the medical domain (ISIC), the mining industry (Ore), and the archaeology domain (OracleScript), as well as the average results across seven datasets on 5-way-1-shot and 5-way-5-shot settings. The numbers in bold indicate the best performance, while the underlined ones denote the second best. All the backbone of the following models is ViT.**

| Method | ISIC | | Ore | | OracleScript | | Average | |
|---|---|---|---|---|---|---|---|---|
| | 1-shot | 5-shot | 1-shot | 5-shot | 1-shot | 5-shot | 1-shot | 5-shot |
| *Unimodality Few-Shot Learning Models* | | | | | | | | |
| FewTURE [CVPR2020] [46] | $33.75_{\pm0.20}$ | $39.23_{\pm0.20}$ | $34.38_{\pm0.35}$ | $44.13_{\pm0.38}$ | $\underline{28.61}_{\pm0.72}$ | $33.09_{\pm0.45}$ | 35.26 | 46.27 |
| HTCTrans [CVPR2022] [20] | $\mathbf{35.76}_{\pm0.46}$ | $49.97_{\pm0.54}$ | $43.76_{\pm0.39}$ | $56.21_{\pm0.44}$ | $28.60_{\pm0.48}$ | $\underline{37.04}_{\pm0.57}$ | $\underline{43.05}$ | 52.45 |
| CPEA [ICCV2023] [19] | $34.95_{\pm0.33}$ | $\underline{50.67}_{\pm0.36}$ | $38.47_{\pm0.41}$ | $59.94_{\pm0.36}$ | $27.10_{\pm0.36}$ | $31.60_{\pm0.38}$ | 40.95 | $\underline{54.96}$ |
| *Vision-Language Models* | | | | | | | | |
| CLIP [ICML2021] [34] | $20.00_{\pm0.55}$ | $20.02_{\pm0.54}$ | $55.35_{\pm0.50}$ | $56.40_{\pm0.54}$ | $19.93_{\pm0.58}$ | $20.03_{\pm0.56}$ | 38.68 | 38.93 |
| Tip-Adapter [ECCV2022] [51] | $20.40_{\pm0.46}$ | $22.20_{\pm0.44}$ | $\underline{57.62}_{\pm0.53}$ | $\underline{61.12}_{\pm0.55}$ | $21.60_{\pm0.53}$ | $26.39_{\pm0.53}$ | 40.13 | 43.67 |
| CoOP [CVPR2022] [52] | $19.80_{\pm0.54}$ | $20.06_{\pm0.58}$ | $55.21_{\pm0.50}$ | $57.34_{\pm0.47}$ | $20.03_{\pm0.52}$ | $20.98_{\pm0.44}$ | 38.54 | 3917 |
| APE-T [ICCV2023] [53] | $20.08_{\pm0.52}$ | $21.74_{\pm0.59}$ | $55.70_{\pm0.49}$ | $59.26_{\pm0.42}$ | $21.57_{\pm0.47}$ | $23.23_{\pm0.44}$ | 40.26 | 42.17 |
| CLIP-Adapter [IJCV2024] [14] | $21.36_{\pm0.79}$ | $22.83_{\pm0.64}$ | $55.74_{\pm0.66}$ | $60.24_{\pm0.63}$ | $20.63_{\pm0.77}$ | $25.47_{\pm0.78}$ | 40.12 | 42.39 |
| MCRNet (Ours) | $\underline{35.25}_{\pm0.44}$ | $\mathbf{52.29}_{\pm0.46}$ | $\mathbf{59.60}_{\pm0.47}$ | $\mathbf{68.95}_{\pm0.48}$ | $\mathbf{29.73}_{\pm0.30}$ | $\mathbf{40.30}_{\pm0.34}$ | $\mathbf{56.11}$ | $\mathbf{68.69}$ |

## 5 EXPERIMENTS

### 5.1 Experiment Setup

**Dataset.** All VLMs were loaded with pre-trained parameters based on CLIP. All visual FSL models were pre-trained on the ILSVRC dataset [8]. It is worth noting that the proposed MCRNet is a CLIP-based model, thus loaded with pre-trained CLIP parameters. During subsequent training, the prototype representation learner, i.e., the CLIP part, was frozen and others were trained on ILSVRC. The reason for not training other VLMs on ILSVRC is that the categories pre-trained by CLIP far surpass those in ILSVRC. Training existing CLIP-based models on ILSVRC did not yield any improvements; in fact, it led to a decline due to catastrophic forgetting. Hence, for an equitable comparison, MCRNet was benchmarked against the top existing methods without relying on VLMs trained on ILSVRC.

All the above methods were evaluated on the seven datasets of the BD-FSIC benchmark mentioned in Sec. 4.

**Testing Strategy.** During the test phase, a standardized $N$-way-$K$-shot approach was used to select support images, with 15 query images sampled per class. The reported results in the tables are presented in both 5-way-1-shot and 5-way-5-shot formats, where 5 novel classes are randomly selected each time, with 1 or 5 support images per class. This constitutes a single-episode test. To ensure fairness, each method underwent 600 random tests. The reported metrics include the average accuracy and a 95% confidence interval. All results are presented as accuracy, representing the proportion of correctly predicted outcomes to the total count.

**Implementation Details.** All models were trained and tested on the same GPU. In the case of MCRNet, the training process was parallelized across eight NVIDIA A800-SXM4-80GB GPUs. After

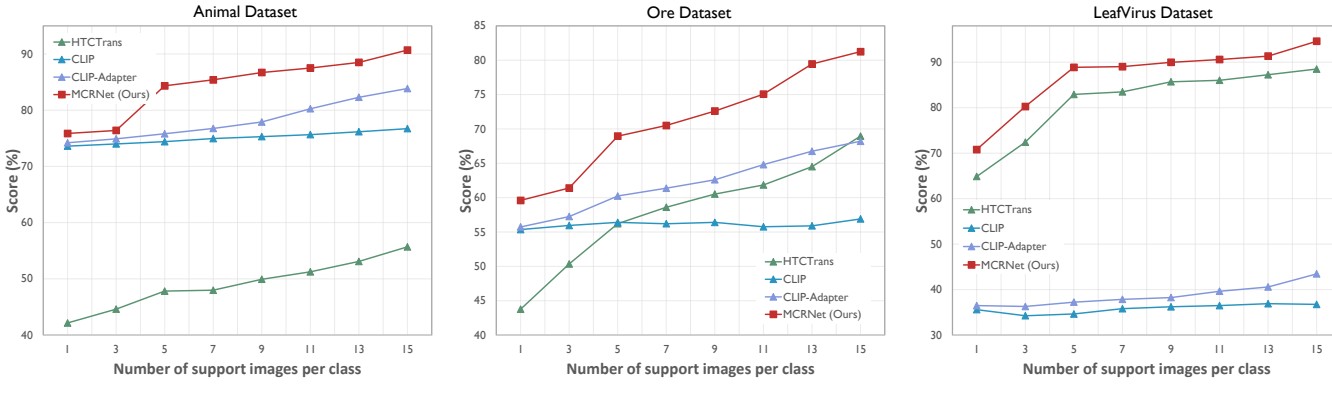

**Figure 4: The performance comparison, as the number of support images increases, among our MCRNet, typical FSL method HTCTrans, the baseline CLIP, and the latest CLIP-based model CLIP-Adapter.**

loading the pre-trained CLIP parameters, both the text and image encoders in CLIP were frozen, and only the other parts were trained on the ILSVRC dataset. The training utilized the Adam optimizer [27] with a learning rate of 0.001. Weight decay was set to 0.05 every 30 epoch, totaling 120 rounds of training. During class-adaptive fine-tuning inference, the collaborative relation learner is frozen, with only the re-representation part undergoing fine-tuning. We employed data augmentation techniques such as rotation and cropping to generate five images per support image, forming 100 image-text triplets for 5-way-5-shot. In the $n$-shot setting, the prototype features of each class's images were averaged to obtain class prototype features for subsequent predictions.

## 5.2   Comparative Experiments and Analysis

We compared MCRNet with SOTA visual FSL models, VLMs, and the baseline CLIP on the BD-FSIC benchmark, as shown in Tab. 1 and 2. Apart from a slight 0.51% lower performance compare to HTCTrans in the 5-way-1-shot setting on the ISIC dataset, MCR-Net outperform all other models in all settings on the remaining datasets. On average across the seven datasets, MCRNet surpass the second-best model by 13.06% in the 1-shot and 13.73% in the 5-shot. Specifically, on the coarse-grained Animal, MCRNet outperform VLMs, particularly surpassing APE-T by 4.36% in the 5-shot setting and the best FSL model by 32.26%. On the Insect dataset, MCRNet's performance is even more remarkable, significantly outperforming existing methods by 22.80% and 22.06%. This demonstrates MCRNet's ability to generalize effectively across different classification granularities, handling scenarios with small intra-class variances. MCRNet maintains a stable advantage on other fine-grained datasets as well, especially on ISIC where its 5-shot results exceed the second-best model. Compared to MCRNet's baseline CLIP, MCRNet show an improvement of 17.43% and 29.76% on average. These results collectively showcase the superiority of our approach, highlighting its strong generalization and practical applicability across multiple domain datasets. Furthermore, we have the following discoveries and analyses:

1) Weaknesses of VLMs: Visual FSL models outperform VLMs on OracleScript and LeafVirus. This demonstrates that existing VLMs

overly rely on the representational abilities learned during CLIP pretraining. When faced with unfamiliar tasks, the textual features in CLIP fail to provide the image encoder with accurate cues, resulting in the image encoder's performance being inferior to that of pretraining models trained solely on visual data. The proposed MCRNet effectively addresses this limitation by incorporating relationship learners that conduct relation learning between support and query images with text. These fused image-text features undergo a re-representation process, correcting the representational biases introduced by CLIP's unfamiliarity. Besides, this also suggests the need to design supplementary representational structures when applying VLMs in specific domains, rather than solely relying on simple adapters or metric enhancements.

2) N-shot inference comparison: We conducted a comparison between the performance of MCRNet and two top-performing models across different $n$-shot scenarios, as in Fig. 4. It is clear that as the amount of provided support data increases, the growth trend of conventional VLMs is not as significant as that of visual FSL models and MCRNet. For example, in the case of 1-shot scenarios, CLIP's accuracy is 4% lower than MCRNet, but by the 15-shot mark, CLIP lags behind MCRNet by almost 25%. Results from CLIP-adapter show a slight improvement but still trail MCRNet by 4% in the 1-shot scenario, surpassing it by more than 15% in the 15-shot scenario. These comparisons underscore that, in contrast to existing VLMs, MCRNet effectively utilizes support image information, swiftly grasping the data distribution of new classes through category-adaptive fine-tuning methods. Furthermore, when compared to FSL models, MCRNet adeptly employs textual cues, maintaining its superiority over them.

3) Dataset analysis: Existing models perform well in general domains, but they show overall poor performance on the BD-FSIC benchmark, especially on the skin disease classification dataset ISIC and the OracleScript dataset for Oracle bone script recognition. Even with an increase in the number of support provided, their performance improvement remains slow. This is because the representation attention of these medical data often focuses on local features such as color and texture, rather than the target shapes



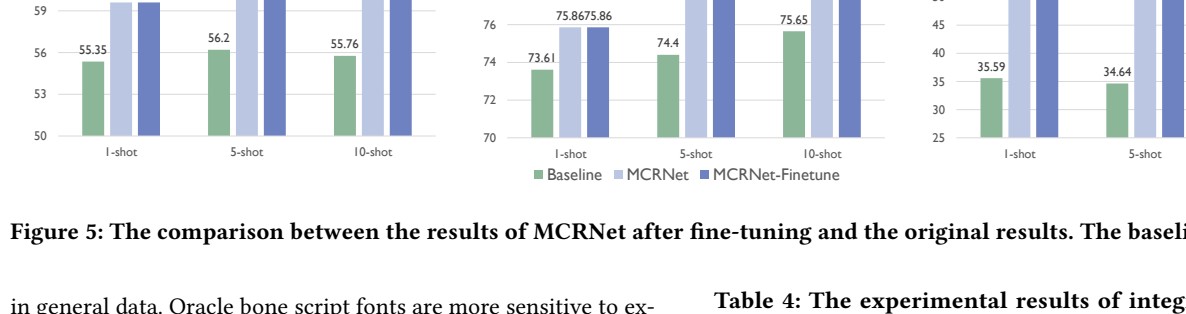

**Figure 5: The comparison between the results of MCRNet after fine-tuning and the original results. The baseline is CLIP [34].**

in general data. Oracle bone script fonts are more sensitive to extracted features because of the significant feature variance within similar fonts. These results indicate the need to pay more attention to the generalization performance of enhancement methods in domain applications to meet practical requirements, while also highlighting the importance of the proposed BD-FSIC benchmark.

## 5.3 Ablation Studies

**Table 3: The impact of the number of iterations on performance in fine-tuning. All results are based on the 5-way-5-shot setting.**

| Iteration(#) | Animal | Ore | LeafVirus | Time(s) |
|---|---|---|---|---|
| 0 | 63.29 | 82.37 | 80.46 | 0.065 |
| 1 | 68.95 | 84.33 | 88.87 | 0.14 |
| 3 | 69.25 | 84.01 | 88.92 | 0.23 |
| 5 | 67.25 | 85.32 | 86.72 | 0.41 |
| 10 | 67.01 | 83.24 | 81.54 | 0.69 |

**The Effectiveness of the Class-adaptive Fine-tuning Inference.** To fully utilize support images and text information, we designed a class-adaptive fine-tuning inference method for MCRNet. In the 5-way-5-shot scenario, 100 image-text pairs were constructed using support images, and so forth. The results in Fig. 5 demonstrate its effectiveness. It can be observed that with five or more support images, this fine-tuning technique consistently boosts performance by 6% on Ore, 2% on Animal, and around 5% on LeafVirus compared to the non-fine-tuned model. The results in the figure are based on a single iteration of constructed data. Tab. 3 illustrates the impact of different numbers of iterations on the results. It is evident that after more than 5 iterations, the model tends to overfit due to the limited data for fine-tuning. Iterating once yields the optimal performance on average with the least time consumption. Therefore, we set the standard number of fine-tuning iterations for MCRNet as one.
**The Flexibility of MCRNet.** MCRNet integrates and re-represents multi-modal information relationships based on prototype features,

**Table 4: The experimental results of integrating MCRNet with FSL models. All results are based on the 5-way-5-shot setting.**

| Method | Animal | Insect | Ore | LeafVirus |
|---|---|---|---|---|
| FewTURE | 44.44 | 44.13 | 44.13 | 74.99 |
| +MCRNet | 55.01 | 58.23 | 46.65 | 76.24 |
| HTCTrans | 47.82 | 59.03 | 56.21 | 82.92 |
| +MCRNet | 56.33 | 64.98 | 59.47 | 83.29 |

making it independent of feature extraction. To demonstrate the flexibility of MCRNet, we integrated it with visual FSL methods, using CLIP's text encoder to extract textual information. As shown in Tab. 4, MCRNet is capable of enhancing textual semantics and improving domain performance on top of visual FSL. Therefore, both visual FSL methods and VLMs can benefit from our work.

## 5.4 Conclusion

To address the under-performance of existing visual few-shot learning models and CLIP-based vision-language models in domain-specific tasks, this paper introduces a novel vision-language collaborative representation network. Building upon CLIP, this network innovatively integrates and collaborates visual and textual features for joint feature fusion and representation, enabling the learning of aligned representations that generalize to new classes. Furthermore, a new evaluation benchmark comprising five domains with seven datasets is proposed to offer a comprehensive domain image classification assessment platform. Comparative experiments demonstrate the superiority and generalization of our approach, with extension experiments showing MCRNet's flexibility in integration with other methods to enhance performance. In the future, we aim to incorporate large language models to enrich existing semantic information, and explore more detailed descriptions to guide multi-domain visual tasks.

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
