# OpenReview forum: "Visual-Language Collaborative Representation Network for Broad-Domain Few-Shot Image Classification"
_acmmm.org/ACMMM/2024/Conference — MM2024 Poster_

### Official Review · Reviewer_mLDc · 2024-05-22

**Rating:** 5
**Confidence:** 2

**Summary:**

The paper introduces MCRNet, a visual-language collaborative representation network designed to improve few-shot image classification by generating category-agnostic relational matrices for better multi-modal feature alignment. Additionally, the authors also introduce a class-adaptive fine-tuning inference technique and a brand-new benchmark. Experiments on this benchmark show that MCRNet outperforms current models, achieving significant improvements in 1-shot and 5-shot settings.

**Strengths:**

1. This paper addresses a fundamental problem in few-shot learning, which is the category-agnostic representation learning. It differs from existing CLIP-based methods in that it introduces an extra step to modeling the collaborative relation between the text, query, and support, enabling more comprehensive semantic alignment. It is not an intricate idea, but personally I think the novelty is acceptable.
2. The authors propose a new benchmark for broad-domain few-shot learning in this work.
3. The experimental design is good, and the results are satisfactory.

**Limitations:**

Major concerns:
1. Besides accuracy, other metrics such as F1 should also be reported in the table.

Minor concerns:
1. Please improve consistency throughout the paper. It seems like "re-presentation" and "re-representation" are chaosly used in the paper.
2. I would suggest the authors add some more intuitive introduction to existing few-shot learning methods if that is possible.

**Suitability:**

3

---

### Official Review · Reviewer_NqKQ · 2024-05-24

**Rating:** 5
**Confidence:** 3

**Summary:**

This paper presents a novel multi-modal collaborative representation network (MCRNet) designed to enhance the alignment and representation capabilities of multi-modal information in unfamiliar domains. Contrary to existing approaches, MCRNet employs a universal relational matrix learning methodology to streamline the fusion and feature representation of multi-modal information. This paper establishes a new benchmark for broad-domain few-shot image classification, comprising seven evaluation datasets from five diverse domains.

**Strengths:**

+This paper is well-written, well-organized, and easy to understand.

+The application scenario of few-shot classification tasks inherently involves arbitrary categories. Addressing CLIP's generalization capability in such scenarios is a very practical issue.

+This paper constructs a meaningful benchmark and proposes an effective method to address the issue.

+This paper conducts extensive experiments to validate the performances of this method under various domains.

**Limitations:**

-In the introduction, the authors mention that CLIP is not adept at handling unfamiliar domains. The authors cite examples of fields such as medicine and industry that are not well-covered by CLIP, and also points out issues with fine-grained classification. Whether these two scenarios require separate discussions and solutions?

-Apart from conducting experiments on domains that CLIP is unfamiliar with, it is recommended for the authors to also report the results on the commonly used datasets. This would further demonstrate the robustness and generalizability of the proposed approach.

**Suitability:**

3

---

### Official Review · Reviewer_wugs · 2024-05-25

**Rating:** 4
**Confidence:** 2

**Summary:**

This work proposes a new method (MCRNet) to integrate the vision and language information for few-shot image classification.
Empirical results demonstrate the effectiveness of MCRNet.

**Strengths:**

1. The paper is well-written and the idea is easy to follow.
2. Extensive experiments are conducted to demonstrate the effectiveness of MCRNet.

**Limitations:**

Some important related works are missing:
[1] deals with the same problems of visual-language integration, but not compared with.
[1] Li, Zhuoling, and Yong Wang. "Better Integrating Vision and Semantics for Improving Few-shot Classification." Proceedings of the 31st ACM International Conference on Multimedia. 2023.

**Suitability:**

2

---

### Official Review · Reviewer_EDk3 · 2024-05-28

**Rating:** 4
**Confidence:** 2

**Summary:**

The paper presents MCRNet, a visual-language collaborative representation network for improving few-shot image classification in specialized fields. MCRNet enhances multi-modal feature alignment through relational matrices and includes visual-text encoders, a collaborative relation learner, and a re-presentation module, plus a class-adaptive fine-tuning technique. A new benchmark with seven datasets from five domains is introduced. Experiments indicate that MCRNet outperforms current state-of-the-art models across various domains.

**Strengths:**

1. The paper establishes a new benchmark that comprehensively covers multiple domains, providing a robust platform for evaluating few-shot learning models.
2. MCRNet significantly outperforms existing state-of-the-art models, achieving notable improvements in both 1-shot and 5-shot settings across various datasets.

**Limitations:**

1. The class-adaptive fine-tuning method might lead to overfitting, especially with limited support data, as indicated by the performance drop after more iterations of fine-tuning.
2. MCRNet relies on pre-trained CLIP parameters, which might limit its applicability if these parameters are not available or if there are significant domain shifts from the pre-training data.
3. While the paper includes fine-grained datasets, further evaluation on more diverse and larger-scale datasets could strengthen the evidence of MCRNet's generalizability.

**Suitability:**

3

---

### Meta-Review · Area_Chair_UMSC · 2024-07-02

**Recommendation:** Accept (Poster)
**Confidence:** 5

**Metareview:**

This paper introduces MCRNet, a novel visual-language collaborative representation network to improve few-shot image classification in specialized fields like healthcare and agriculture. Unlike existing CLIP-based models, MCRNet generates relational matrices for multi-modal feature alignment, which is category-agnostic and generalizes to new domains. It also employs class adaptive fine-tuning for better learning with limited data. The new broad-domain few-shot benchmark shows MCRNet's superior performance, achieving significant improvements over state-of-the-art models.

Pros
- Establishes a comprehensive benchmark covering multiple domains, providing a robust platform for evaluating few-shot learning models.
- MCRNet significantly outperforms existing state-of-the-art models, with notable improvements in 1-shot and 5-shot settings.
- The paper is well-written, well-organized, and easy to follow.
- Conducts thorough experiments to validate MCRNet's effectiveness across various domains.
- Addresses the practical issue of CLIP's generalization capability in few-shot classification tasks involving arbitrary categories.
- Introduces a novel approach for category-agnostic representation learning, enhancing semantic alignment through collaborative relation modeling.

Cons
- The class-adaptive fine-tuning method might lead to overfitting with limited support data, as indicated by performance drops after more iterations.
- MCRNet relies on pre-trained CLIP parameters, which may limit applicability if these parameters are unavailable or if there are significant domain shifts.
- While the paper includes fine-grained datasets, further evaluation on more diverse and larger-scale datasets could strengthen the evidence of MCRNet's generalizability.
- Important related works, such as Li and Wang (2023), which address similar problems of visual-language integration, are not compared.
- The paper mentions issues with CLIP handling unfamiliar domains and fine-grained classification but does not provide separate discussions or solutions for these scenarios.
- The paper should include results on commonly used datasets to further demonstrate the robustness and generalizability of MCRNet.

The rebuttal has addressed most of the cons from reviewers, and all reviewers agree this paper should be accepted.